Combinations of action observation and motor imagery on golf putting’s performance

Lin Chi-Hsian 1
Lu Frank J.H. frankjlu@gmail.com 2
Gill Diane L. 3
Huang Ken Shih-Kuei 2
Wu Shu-Ching 4
Chiu Yi-Hsiang 2
1 Physical Education Office, National Taipei University , Taipei City , Taiwan
2 Department of Physical Education, Chinese Culture University , Taipei , Taiwan
3 Department of Kinesiology, University of North Carolina at Greensboro , Greensboro , NC , United States of America
4 Center for General Education, Ling-Tung University , Taichung , Taiwan
Cè Emiliano
Electronic publication date: 2022 May 11
Publication date: 2022
Volume: 10
Electronic Location ID: e13432
Received 2021 Nov 30; Accepted 2022 Apr 22
Copyright: ©2022 Lin et al.
Copyright year: 2022
Copyright holder: Lin et al.
License: This is an open access article distributed under the terms of the Creative Commons Attribution License, which permits unrestricted use, distribution, reproduction and adaptation in any medium and for any purpose provided that it is properly attributed. For attribution, the original author(s), title, publication source (PeerJ) and either DOI or URL of the article must be cited.
License URL: https://creativecommons.org/licenses/by/4.0/

Keywords: Motor skill, Cognitive process, Mental practice, Mental simulation

Funding: The Ministry of Science and Technology (MOST) of Taiwan research grant MOST 108-2410-H-305-067- The Ministry of Science and Technology (MOST) of Taiwan research grant to professor Chi-Hsian Lin (MOST 108-2410-H-305-067-) supported this work. The funders had no role in study design, data collection and analysis, decision to publish, or preparation of the manuscript.

==============================
Motor imagery (MI) and action observation (AO) have been found to enhance motor performance, but recent research found that a combination of action observation and motor imagery (AOMI) together is even better. Despite this initial finding, the most effective way to combine them is unknown. The present study examined the effects of synchronized (i e., concurrently doing AO and MI), asynchronised (i.e., first doing AO then MI), and progressive (first asynchronised approach, then doing synchronized approach) AOMI on golf putting performance and learning. We recruited 45 university students (Mage = 20.18 + 1.32 years; males = 23, females = 22) and randomly assigned them into the following four groups: synchronized group (S-AOMI), asynchronised group (A-AOMI), progressive group (A-S-AOMI), and a control group with a pre-post research design. Participants engaged in a 6-week (three times/per-week) intervention, plus two retention tests. A two-way (group × time) mixed ANOVA statistical analysis found that the three experimental groups performed better than the control group after intervention. However, we found progressive and asynchronised had better golf putting scores than synchronized group and the control group on the retention tests. Our results advance knowledge in AOMI research, but it needs more research to reveal the best way of combining AOMI in the future. Theoretical implications, limitations, applications, and future suggestions are also discussed.

Introduction

Imagery is a widely used psychological skill that uses all senses (i.e., visual, olfactory, auditory, kinesthetic, gustatory) to create or recreate an experience in mind without overt behavior (Jeannerod, 1994; Vealey & Forlenza, 2015). Theoretically, imagery is “a simulation representation of motor behavior; it can be seen as either a class of an inferred cognitive structure or processes or a class of more or less perceptual-like experience that happens in mind (Richardson, 2013; p.3)”. Today, many terms are used for imagery, such as visualization, mental practice, mental imagery, mental rehearsal, and covert practice (Debarnot et al., 2014). To further understand how imagery works and how it influences performers’ behavior, researchers have proposed several theories such as Jacobson’s (1931) psychoneuromuscular theory, Sackett’s (1934) symbolic learning theory, Lang’s (1977, 1979) bio-informational theory, and Jeannerod’s (1994) functional equivalence hypothesis to explain the mechanism of imagery. Each theory has received empirical studies’ support in the past few decades.

For example, to examine athletes’ psychophysiological responses during motivational imagery scenarios, Cumming, Olphin & Law (2007) sampled 40 competitive athletes wearing a standard heart rate monitor and asked them to imagine five scripts with mastery, anxiety, coping, psyching-up, and relaxation. Results found participants’ heart rate increased after they imaged anxiety, coping, and psyching-up imagery scripts. The imagery-induced effects supported Lang’s (1977, 1979) hypothesis that images containing response propositions will produce a physiological response. Similarly, in a study that examined whether imagery could influence athletes’ appraisal of stress-evoking situations and whether cardiovascular responses varied according to the cognitive appraisal of imagery scripts, Williams, Cumming & Balanos (2010) sampled 20 athletes to imagine three scripts with the challenge, neutral, and threat; and measure their heart rate, stroke volume, and cardiac output during imagery. Results found using different types of imagery scripts influenced participants’ heart rate, stroke volume, and output changes. Williams, Cumming & Balanos (2010) study not only supports using imagery can facilitate adaptive stress appraisal but also supports Lang’s (1979) assumption that responses will reflect the actual situation. These two examples illustrated that although without overt physical engagement, inner mental simulation of life experiences influences psychophysiological responses and behavior.

Similar to motor imagery, action observation is another type of motor simulation (Eaves et al., 2016). Action observation involves a structured examination of motor acts that can be performed either onsite or using video films/images (Neuman & Gray, 2013). Unlike imagery, action observation does not require generating and maintaining motor representation to control image quality (e.g., clarity, fidelity, perspective) and ability (Holmes & Calmels, 2008). Action observation is believed to promote motor learning and performance because it increases the unconscious activation of motor codes (Giacomo et al., 2021). Early neurophysiological research revealed that when performing an action observation the mirror neuron system in the monkey’s brain (e.g., Di Pellegrino et al., 1992; Rizzolatti et al., 1996) and in the homologous areas of the human brain (e.g., Buccino et al., 2001; Mukamel et al., 2010; Thill et al., 2013) has been activated. Some researchers considered (e.g., Rizzolatti, Fogassi & Gallese, 2001) that during observation of a movement, the related action representation will be re-activated in the motor system. Rizzolatti, Fogassi & Gallese (2001) called this process is “motor resonance” which can drive learning through a facilitatory effect on motor pathways (Buccino et al., 2001; Wheaton et al., 2004). Caspers et al. (2010) contended that the cortical areas stimulated in action observation correspond to those that are activated in the actual movement. Furthermore, it was indicated that the activation process in the brain of action observation is similar to the imagery process (Munzert et al., 2008), and action observation can replicate the kinematic characteristics (e.g., speed) displayed in demonstrations as observers generate movements’ biological actions through the lower-level mechanisms of the action observation network (Wild et al., 2010).

Motor imagery and action observation are conventionally regarded as two different techniques, and researchers often compare their effects on motor performance separately (e.g., Ram et al., 2007; Neuman & Gray, 2013). However, it is suggested that combing MI and AO together (i.e., AOMI) is even better than practicing them separately (Eaves et al., 2016; Vogt et al., 2013; Wright, Frank & Bruton, 2021). The general procedure to perform an AOMI intervention is to ask participants to observe the actions presented in videos and imagine the physiological sensations/ behavioral responses either synchronisedly or serially (e.g., Romano-Smith, Wright & Wakefield, 2018; Scott et al., 2018; Sun et al., 2016). By such an approach, it is contended that participants will gain visual information of a novel motor skill during action observation or kinesthetic feeling in muscles/ joints during motor imagery (Wright, Frank & Bruton, 2021). Further, either visual information regarding movement technique during action observation or kinesthetic feelings during motor imagery enable participants to focus their attention toward movement execution (Eaves et al., 2016).

Empirical studies confirm that AOMI intervention increases muscle strength, (Scott et al., 2018; Wright & Smith, 2009), balance (Taube et al., 2014), golf putting (Smith & Holmes, 2004), and dart throwing accuracy (Romano-Smith, Wright & Wakefield, 2018). In neurophysiological research, it was found that AOMI increases the activation of motor cortical areas in the brain (Wright et al., 2018).

Despite these initial findings, the optimal order of action observation and imagery remained unknown. Many studies focus on the effectiveness of AOMI on rehabilitation or examine the cortical activities of AOMI (e.g., Bruton et al., 2020; Castro et al., 2021; Emerson et al., 2022), but what is the effective way to combine AO and MI together has been rarely examined. For example, in a systematic review and meta-analysis that examined the effects of AOMI on pain intensity of patients with musculoskeletal pain, Suso-Martí et al. (2020) found compared to traditional rehabilitation, AO and MI significantly reduced pain intensity. Similarly, Herranz-Gómez et al. (2020) conducted an umbrella and mapping review with meta-meta-analysis to examine the effectiveness of MI and AO on arm functionality and performance in stroke patients. Results found AO and MI intervention improved patients’ arms’ functionality and performance. In a narrative study, Wright, Frank & Bruton (2021) provided an overview of the literature of AOMI and discussed the neurophysiological, cognitive, psychological, and performance effects of AOMI interventions. They proposed several practical recommendations for practitioners on how to develop and implement AOMI interventions for performance enhancement. However, the optimal way to combine AO and MI remained untouched.

Fortunately, researchers noticed the void part and started to examine the optimal combination of AOMI in enhancing motor performance. For example, Sun et al. (2016) recruited 10 heart stroke patients with right-sided upper extremity hemiplegia to examine whether the combination of motor imagery and action observation improve their upper limb function. The experimental group used a synchronized AOMI (S-AOMI) approach while the control group used asynchronised AOMI (A-AOMI). Sun et al. (2016, p.2) explained that both motor observation and imagery provide related information that enables participants formulated a mental representation in mind which activates neural activities in the brain, and subsequently enhance participants’ abilities to rehearse the movements. Thus, either with synchronized AOMI or asynchronised AOMI can be beneficial to motor performance and learning. As expected, results found the S-AOMI group exhibited significant improvements in the upper limb’s strength and function after 4 weeks of intervention but not A-AOMI. Recently, Romano-Smith, Wright & Wakefield (2018) replicated their study by recruiting 50 university students to perform a dart-throwing. They argued that Sun et al. (2016) sampled heart stroke patients as participants, and muscle training was too limited to be generalized in the complex motor task that requires a high level of coordination and accuracy. They recruited 50 university students and assigned them into the following five groups: S-AOMI, A-AOMI, motor imagery, action observation, and control group to test their effects on dart-throwing. After 6 weeks of intervention three times per week, they found all groups improved significantly except for the action observation and control groups. Notably, both S-AOMI and A-AOMI performed better than the action observation and control group with no differences between S-AOMI and A-AOMI. Romano-Smith, Wright & Wakefield’s (2018) study brings new insights and raises questions about whether S-AOMI or A-AOMI is effective with other motor tasks and participants. However, Romano-Smith, Wright & Wakefield’s (2018) used a throwing motor task (i.e., dart-throwing) as an experimental task. The throwing motor task requires performers to generate a motor program in mind before motor execution (Schmidt et al., 2019) but golf putting is more complicated. Performers need not only to generate a motor program in mind but also has to detect the performer’s stance, rotation of the trunk, swing angle, the contacting surface of the club, and putting force to make a correct movement execution. Thus, whether Romano-Smith, Wright & Wakefield’s (2018) study can be generalized in complex motor skills as golf putting is worthy of investigation.

Present study

We intended to conduct a study that sampled university students as participants to examine how different combinations of AO and MI influence golf putting performance. As earlier stated, the procedures to perform A-AOMI and S-AOMI are different. For A-AOMI, it performs AO first then MI. This approach has several advantages for performers. First, it allows participants to learn AO and MI step by step. Participants can understand the whole picture of the movement, especially, the body position, stance, limbs’ movements, and the process to execute it. In contrast, S-AOMI needs participants to simultaneously perform AO and MI at the same time. This arrangement has several limitations. First, it would confuse performers about which part (AO or MI) should do the first, or how many portions they should practice for each part. They might focus on AO most of the time, then do the MI, or vice versa.

Further, to extend the past research paradigm, we created a progressive AOMI (i.e., A- S-AOMI) by engaging in A-AOMI intervention first, then followed with S-AOMI. Two reasons to use a progressive AOMI. First, we considered participants who have no experience with AO and MI. If we arrange for participants to have the opportunity to learn each skill first, it would be very easy for them to do AO and MI simultaneously. Further, recent neurophysiological research found the A-AOMI approach is less demanding in cognitive processing than S-AOMI (Emerson et al., 2022; Eaves et al., 2016). Thus, by engaging in A-AOMI first, they will gain a basic foundation for AOMI intervention. We expected this arrangement will produce better results. Moreover, because past AOMI combination research only examined the immediate effect on performance, we also examined performance over retention tests. We proposed the following hypotheses:

H1: The S-AOMI, A-AOMI, and A-S-AOMI interventions will perform better than the control group on golf putting.

H2: The A-S-AOMI will perform better than S-AOMI and A-AOMI on golf putting performance.

Materials and Methods

Participants

We used G*Power 3.1.9 with a medium effect size (f = .25), α = .05, and statistical power of 90% to determine our sample size, and found that each group must contain a minimum of 11 participants (i.e., n = 44). Thus, we recruited 48 university students with no golf putting and motor imagery experiences in our study. The final sample size was 45 (Mage = 20.18 ± 1.32 years; males = 23, females = 22). However, three participants withdrew during experiments. Thus, all groups had 11 participants except the second group with 12.

Measures

The revised Chinese Version of the Movement Motor imagery Questionnaire-Revised (MIQ-R, Lin, 2012)-The MIQ-R was used to assess participants’ motor imagery ability of four basic movements including knee lift, jump, arm movement, and waist bend in visual and kinesthetic modalities. Participants are asked to read through each statement and perform the movement described. Participants rate the ease or difficulty of imaging the movement on a 7-point Likert scale ranging from 1 (very hard to see/feel) to 7 (very easy to see/feel). A higher score represents a greater motor imagery ability. Lin (2011) reported that adequate internal reliability of the revised Chinese MIQ-R by composite reliability (CR) - visual motor imagery = .88, and kinesthetic motor imagery = .82; and average variance extracted (AVE) -AVE: visual motor imagery = .65, and kinesthetic motor imagery = .53.

Procedures and tasks

Ethical Approval and Participants Recruitment: Prior to data collection, we got ethical approval from Ethical Review Board (REC), National Taiwan University (NTU-REC No. 201903ES019). Then, we used a bulletin notice and online announcement to target students in a university (excluding golf players). The notice provided information about the study and indicated that the experiment consists of approximately 20-min sessions held three times a week over 6 weeks. Interested participants contacted the first author individually, and were introduced to the purpose and process of the experiment. After he/she understood the experiment and agreed to participate in the study, he/she signed the consent form and followed the experimental arrangements. On the first day of the meeting, all participants filled the MIQ-R to assess their motor imagery abilities. We used this information and randomly assigned participants to different experimental groups.

Intervention

Education phase: In this stage, we introduced participants to the concept and use of AOMI. For motor imagery, the participants were instructed to imagine that they were holding a golf club in a standing position and preparing to putt. Subsequently, they were to focus on details and scenarios related to putting and imagining all emotional and physiological sensations. We used the principles of the physical–environmental–task–timing–learning–emotional–perspective (PETTLEP, (Holmes & Collins, 2001)) approach to teaching motor imagery. Holmes & Collins’s (2001) functional equivalence hypothesis contends that imagery is a cognitive process that activates the brain to prepare, plan, and execute the movement in the mind before an overt motor performance, and is equivalent to actual motor performance (Holmes & Collins, 2001, p. 62). While doing the PETTLEP imagery, performers have to integrate all the elements of an actual motor performance including physical condition, environmental settings, motor task, timing, learning attitude, emotional states, and imagery perspective to the mental simulation process. Therefore, we used PETTLEP approach to practice MI.

For action observation, we used six golf putting videos taken from the first-person perspective by a professional golfer. Each video included five successful putts. The first video spanned approximately 2.13 min, with a demonstration in conjunction with a voice explaining the key points to performing the action. The subsequent five videos (1 min each) showed five successful putts (12 s each). A question-and-answer session was conducted after the videos were played, followed by a simple instructional session on golf putting skills.

Pretest: After participants finished the education session, they completed the Chinese MIQ-R (Lin, 2011) to assess their visual and kinesthetic imagery ability. Next, they practiced golf putting three trials (five putts each) for a warm-up (Smith & Holmes, 2004). Then, participants performed the golf putting skill test with six trials (five putts each), for a total of 30 putts.

Stimulus-Response Training: After the pretest, the experimental groups received stimulus–response training following the bioinformational theory proposed by Lang et al. (1980). According to Lang et al. (1980), when performing imagery, performers establish several well-organized statements including stimulus statements and response-statement. The stimulus statement refers to the description of those stimuli in the environments such as basketball court, audience, ball, teammates, and opponents. On the other hand, a response statement is how athletes/performers react to these stimuli including physiological and kinesthetic responses. The appropriate stimulus–response connection allows learners to perform the correct action after imagery training. To apply this suggestion to our study, we arranged participants to experience specific stimuli (e.g., details regarding the environment) and responses (e.g., physiological sensations such as muscle tension), visceral events (e.g., increased heart rate), and sensory adjustments (e.g., postural changes) in the imagery process. We used a pre-recorded video that contained response statements to produce vivid imagery for participants (Williams, Cooley & Cumming, 2013). We asked participants to reconfirm the imagery contents by intervention questionnaire to make sure they understand the imagery process. Before the formal experiment, participants were instructed to engage in imagery with either eye opened or closed, adding details to strengthen their imageries. By doing so, the participants generated their content rather than using researcher-provided imagery scripts. Moreover, they were free to adjust the imagery content as they wished. Three stimulus–response training sessions were conducted thrice weekly for 2 weeks (six sessions total) to ensure that the participants had sufficient opportunity to accept relevant stimulus statements and produce responses that met their individual needs, regardless of the specific intervention.

Intervention Execution: The three types of intervention were as follow:

(a) S-AOMI group (synchronised-AOMI): To perform an S-AOMI intervention, the participants (males =4; females =7) engaged in six sections of intervention. First, they watched a professional golfer’s putting and simultaneously imaged the head and trunk position and rotation, the kinesthetic sensation of the putting, and visual trajectory of the ball route until the ball rolled into the hole (for detail, please refer to the imagery script in supplement). Each section was proceeded by five trials (i.e., each trial should simultaneously do action observation and motor imagery together). Then, they had 30 s break. Next, they performed another section of intervention until all six sections of intervention were completed. The total number of AOMI golf putting interventions was 30. Using this approach, they engaged in 18 intervention sessions held three times weekly over 6 weeks.

(b) A-AOMI group (asynchronised-AOMI): First, participants (males = 6; females = 6) watched the golf putting video as previously stated with five successful putts. They observed the videos carefully and memorized all components of successful putting. Then, they imaged successful golf putting by using the principles of PETTLEP approach. Next, they had 30 s break and went to another section of intervention as aforementioned procedure. Because A-AOMI started five trials of AO plus five trials of MI, they only engaged in 3 sections of intervention which accumulated to 30 golf putts in total.

(c) A-S-AOMI group (progressive-AOMI): First, participants (males = 7; females = 4) engaged in A-AOMI (i.e., five AO first, then five MI which made 10 golf putting), then they engage in S-AOMI which made five golf putts. Then, they repeated A-AOMI and S-AOMI until all 30 golf putts were completed. The detailed procedures of two types of AOMI are as previously stated.

(d) Control group: During the experimental periods, the control group (males = 6; females = 5) spent the same amount of time reading a story about a celebrity golfer. The content did not include mental training or techniques for improving physical skills. This design is similar to previous studies (e.g., Smith & Holmes, 2004; Smith, Wright & Cantwell, 2008).

Post-test: After the interventions, all participants underwent a pos t-test. First, they performed three trials (five putts each) for a warm-up. Then, they performed six trials (five putts each) a formal pos t-test which makes a total of 30 putts.

Retention test: We adopted Jennings, Reaburn & Rynne (2013) suggestion to assess participants’ golf putting performance by 10 min after the final intervention and 1 week after the posttest (i.e., three trials for a warm-up and six trials for formal test, and each trial with five putts). The formal test for retention test was 30 putts.

General Information about experiment

The experiments were conducted in a university indoor golf putting area, and the control group received their treatments in a reading room beside a golf putting area. To avoid additional learning effects, we asked participants not to observe golf putting or image the golf putting during their free hours. Also, after each intervention, the experimental groups were asked to complete intervention questionnaires. The intervention questionnaires were used to encourage the participants to actively engage in each task, as well as to confirm that they were following the guidance they were given (Marshall & Wright, 2016; Williams, Cooley & Cumming, 2013).

Research Tools: We used the following tools to conduct our experiments.

(a) MP3 players: Relevant instructions were played on MP3 players during the stimulus–response training and initial imagery guidance, the participants used the MP3 player to modify their imagery scripts.

(b) DVD players and video projector: We used DVD players and video projector to present a professional golfer’s golf putting during experiments.

(c) Imagery instructions: The imagery instructions, which were based on the results of the individual interviews which centered on the performance of cognitive-specific imagery. They included the simulated putting preparation, movements during the execution, the path of the golf ball after putting, and the ball’s entry into the hole. In addition to visual guidance, kinesthetic characteristics (classified as implicit stimuli, responses, and statements of the meaning) were considered. These included stance, grip, the positions of the hands and feet during putting, the sound of the ball rolling, and the participants’ emotional reaction after the entry of the ball into the hole. Guidance was provided according to the participants’ preferred imagery perspectives, and the finalization of the imagery content was determined according to the individual needs.

(d) The Chinese version of the MIQ-R: The Chinese version of the MIQ-R was used to assess participants’ motor imagery ability.

(e) Intervention questionnaires: The questionnaire items asked participants’ perceptions of the interventions, such as their level of concentration, the speed at which they generated the images, and the clarity of the images. Six items were scored on a 7-point Likert scale (with 1 and 7 indicating strongly disagree and strongly agree, respectively). They are presented as follows:

• Item 1: I am focused during practice.

• Item 2: I follow the script during imagery practice.

• Item 3: My actual putting movement and performance correspond to those simulated imagery training or action observation.

• Item 4: I follow the actual putting speed during imagery practice.

• Item 5: The images I generate during imagery practice (or action observation) are clear.

• Item 6: My senses (e.g., touch and kinesthesia) are engaged in the images I generate.

• Item 7: The intervention is effective to me.

(f) Golf putting equipment: The golf putting equipment included artificial golf turf, target holes, golf clubs, golf balls, and devices for distance measurement (see Fig. 1).

(g) Skill performance scoring: A target hole 10 cm in diameter was placed on a section of artificial turf with 180 cm in length and 90 cm in width. This skill scoring system is similar to previous studies (e.g., Beilock & Gonso, 2008; Ismail, 2014; Ismail, 2015; Smith & Holmes, 2004). Before experiments, we consulted a golf expert and he suggested that 180 cm is an ideal distance for the novice. Then, we invited 3 participants to perform it and found this distance is appropriate neither too difficult nor too easy. Further, to reduce variation of the golf putting, we used a flat artificial turf for the experiments. The starting point of the putter was marked with a sticker 180 cm from the target hole (Fig. 1). For each of the 30 putts performed over one intervention session, five points were awarded if the ball directly entered the hole, three points were awarded if the ball passed by and touched the edge of the hole without entering it, two points were awarded if the ball passed by the hole without touching it because they applied the right putting force but failed to control direction, and 1 point was awarded if the golf ball did not reach the hole because they neither control putting force nor putting direction. (Ismail, 2014; Ismail, 2015).

(h) Experimental control: To prevent participants from doing extra training and engagement in golf putting, we asked them not to observe or practice golf putting during the experiment period. All participants agreed to follow this control and reported that they didn’t engage in the aforementioned behavior. The general procedures to complete the experiment are illustrated as Fig. 2.

Statistical analyses

Before formal statistical analyses, we screened all data by examining means, standard deviations, skewness, kurtosis, and outliers to make sure there were no abnormal data. Further, we used descriptive statistics to examine the participants’ responses to the intervention questionnaire. To assess the effects of various AOMI on golf putting learning and performance, we used a two-way mixed-design analysis of variance (4 groups ×3 times) to examine between-group differences across various assessments. Tests of simple main effects were followed up when significant interaction effects appeared. The effect size was computed and reported as a partial η2 value. When there is a significant difference among groups, we used the least significant difference (LSD) test to perform post-hoc differences. The significance level of the present study was set at an alpha level of .05 prior to LSD. However, to provide more information about the alpha value we also presented a detailed magnitude of all alpha values.

Figure 1 Overhead view of the experiment set-up.

This figure illustrates how the experiment was performed.

Figure 2 The flowchart of the experimental process.

This figure shows how experiment was proceeded.

Results

Manipulation check

According to intervention questionnaires, participants were able to follow our instructions. Table 1 shows all groups’ mean scores on seven items ranging between 5.49 and 6.18. This indicates that all participants were performing AOMI for the first time, displayed a high level of adherence to the intervention protocol, felt positively toward the guidance they were given, and perceived the interventions as effective. Also, we used one-way ANOVA to examine the three experimental groups’ imagery ability and found no significant between-group differences in visual, F (2, 31) = 1.21, p > .05, partial η2 = .07, or kinesthetic imagery, F (2, 31) = .87, p > .05, partial η2 = .05.

Effects of different AOMI on golf putting learning and performance

Table 2 presents the descriptive statistics of golf putting scores on the pretest, posttest, and second skill retention assessment. The two-way mixed-design ANOVA revealed that there is no significant difference across four groups in pre-test F(3,44) = 1.13, p = .348. ηp2 = .076). But there is a significant main effects for tests F(3, 123) = 34.63, p < .05, partial η2 = .46 and for groups, F(3, 123) = 4.12, p < .05, partial η2 = .23 except control group (see Table 3). Further, a significant group × test interaction was found F (9, 123) = 4.35, p < .05, partial η2 = .24. Because more than two groups were employed in this study, we used LSD post hoc tests to compare the mean scores of S-AOMI, A-AOMI, and A-S-AOMI and the control group. As Table 3 and Fig. 2 illustrate, all experimental groups scored significantly higher than the control group, and A-S-AOMI was significantly higher than S-AOMI and A-AOMI at the posttest. Further, at the first retention test, the A-AOMI and A-S-AOMI groups scored significantly higher than the control group, and the A-AOMI scored significantly higher than the S-AOMI group. Moreover, at the second retention test, A-AOMI and A-S-AOMI were significantly higher than the control group, and the A-S-AOMI was significantly higher than the S-AOMI group (see Table 3 and Fig. 3).

Table 1 Descriptive statistics of intervention questionnaire in three experimental groups.

Groups/items	1 (M/SD)	2 (M/SD)	3 (M/SD)	4 (M/SD)	5 (M/SD)	6 (M/SD)	7 (M/SD)	
S-AOMI	5.91(1.02)	6.14(0.87)	5.68(0.51)	5.63(0.90)	6.41(0.63)	5.95(0.57)	5.45(0.76)	
A-AOMI	6.21(0.75)	6.29(1.01)	5.92(0.70)	5.79(1.16)	5.67(1.40)	5.63(1.30)	5.38(0.91)	
A-S-AOMI	5.91(1.09)	6.09(0.80)	5.77(0.60)	5.45(1.17)	5.91(0.94)	5.72(0.93)	5.64(0.95)	

Table 2 Descriptive statistics of golf putting by group and time.

Stage	Pretest	Posttest	Retention 1	Retention 2	
Groups	M	SD	M	SD	M	SD	M	SD	
S-AOMI
(n = 11)	76.36	5.59	89.82	5.98	84.73	6.44	85.27	7.09	
A-AOMI
(n = 12)	82.75	9.03	96.42	13.26	94.92	14.56	93.17	13.31	
A-S-AOMI
(n = 11)	81.55	5.34	99.82	8.23	94.46	11.37	94.55	8.55	
Control
(n = 11)	80.64	12.87	79.91	14.45	81.82	13.85	82.09	9.46	
Notes.

AOMI action observation and motor imagery

S-AOMI synchronized AOMI

A-AOMI asynchronized AOMI

A-S-AOMI asynchronized followed by synchronized AOMI

Table 3 Post-hoc comparisons of golf putting in different groups and tests.

Source	SS	df	MS	F value	η p 2	ρ	Post hoc	
Comparison								
Groups (A)								
Pretest (b1)	260.51	3	86.84	1.13	.08	.348		
Posttest (b2)	2559.70	3	853.23	6.80*	.33	.001	a3>a1; a1>a4; a2>a4; a3>a4	
Retention 1 (b3)	1514.98	3	504.99	3.48*	.20	.024	a2>a1; a2>a4; a3>a4	
Retention 2 (b4)	1223.72	3	407.91	4.10 *	.23	.012	a3>a1; a2>a4; a3>a4	
Residual	18242.09	123	147.50					
Tests (B)								
S-AOMI (a1)	1037.36	3	347.79	50.57*	.84	<.001	b4>b1; b3>b1; b2>b1; b2>b3; b2>b4	
A-AOMI (a2)	1377.56	3	459.19	21.31 *	.66	<.001	b4>b1; b3>b1; b2>b1; b2>b4	
A-S-AOMI (a3)	1996.82	3	665.61	29.62*	.75	<.001	b4>b1; b3>b1; b2>b1; b2>b3; b2>b4	
Control group (a4)	34.43	3	11.48	.16	.02	.924	
Residual 3750.32 123 30.49					
Notes.

* ρ< .05 ; a1: S-AOMI group; a2: A-AOMI group; a3: A-S-AOMI group; a4: control group; b1: pretest; b2: posttest; b3: first skill retention assessment; b4: second skill retention assessment.

Figure 3 The mean scores of the golf putting scores in each experimental group across three assessements.

This figure illustrates the differences of four experimental groups in pre-post test and retention.

Discussion

The main purpose of this study was to extend Romano-Smith, Wright & Wakefield (2018) study by comparing a synthesized A-S-AOMI intervention with traditional S-AOMI and A-AOMI on golf putting performance. The results indicated that all the experimental groups performed better on the posttest than the pretest, but not the control group. Also, the posttest scores of the experimental groups were higher than the control group. Thus, H1 is supported. Further, we compared the posttest scores of all experimental groups and found that the A-S-AOMI had the highest score and was better than S-AOMI but not A-AOMI. Thus, H2 is partially supported. Furthermore, in the first retention test, the A-AOMI group scored higher than S-AOMI and control groups, and A-S-AOMI and S-AOMI scored higher than the control group. In the second retention test, the A-S-AOMI scored higher than S-AOMI and control groups, and A-AOMI and S-AOMI scored higher than the control group. The mixed results on golf putting performance have several implications.

Traditionally, sport psychologists tend to adopt AO and MI as a psychological intervention to enhance athletes’ performance but with a separate manner (please refer to Ste-Marie et al., 2012 for AO, and Mizuguchi et al., 2012 for MI). However, the optimal way to combine AO and MI together remained unknown. The combination of AO and MI has several advantages for motor learning and performance. From the motor learning perspective, clearly understanding the structure of motor skill and the performance routine increase the content knowledge of motor skill (Schmidt et al., 2019). To gain a better learning effect, AO can provide a substantial foundation for motor skills. On the other hand, MI intervention strengthens performers’ kinesthetic perceptions of motor skills. The combination of AOMI brings these two approaches together can foster learning effectively and efficiently. Recent neurophysiological research (e.g., Eaves et al., 2016; Emerson et al., 2022) showed that when combining AOMI, it activates more motor cortical activities than pure AO or MI. The combination of AOMI allows the motor cortex to have a better function to plan, control, and execute the voluntary movement.

Thus, the effects of three types of AOMI on golf putting performance not only support past research (McNeill et al., 2020; Romano-Smith, Wright & Wakefield, 2018; Romano-Smith et al., 2019; Sun et al., 2016) findings that combining imagery with action observation can enhance motor performance but also provides another evidence of the benefits of AOMI. Notably, our study and Romano-Smith, Wright & Wakefield (2018) found that A-S-AOMI and A-AOMI had similar effects on motor tasks while Sun et al. (2016) found S-AOMI performed better than A-AOMI. The difference may be attributed to participants’ characteristics and motor tasks. Sun et al. (2016) sampled heart stroke patients with right-sided upper extremity hemiplegia and performed muscle strength. Heart stroke patients have been found to have problems with working memory (e.g., Constantinidis & Klingberg, 2016). The S-AOMI approach has only one step to engage in the intervention so it has the advantage to reduce the demand for working memory. In contrast, the A-AOMI intervention needs to engage in action observation first, then succeeded with motor imagery. Due to working memory, it might undermine the effectiveness of A-AOMI on motor performance. We suggest future studies may compare the effects of different combinations of AOMI on motor performance with different populations.

As to the nature of the motor task, Sun et al. (2016) asked heart stoke participants to observe a model uses his/her right arm to insert a peg on a hole of a wooden board and then remove it from the board either by imaging it synchronously or asynchronously (Sun et al., p.3). Such a simple motor task requires fewer cognitive processes. Golf putting requires a more complex mental process. According to Wang et al. (2020), when performing golf putting, performers have to block competing attention, focus on the contacting surface of the club’s head, estimate putting difference, and exert a proper force along the designated route to the hole. These complex cognitive processes generally need the involvement of motor cortical areas such as the right motor cortex and left/medial cerebellum (Nyberg et al., 2006). We suggested that future studies may compare the effects of a different combination of AOMI on different motor tasks. Further, because we did not measure brain physiological activity, we suggest future studies compare motor cortical activities in different AOMI combinations.

The combination of AO and MI has its theoretical implication for the researchers. As earlier stated, past research tends to separate AO from MI and examine their effects on motor skills performance. Recent neuroimaging studies found that a combination of AOMI increases motor cortical activities in the brain than only doing AO or MI (Holmes & Wright, 2017; Eaves et al., 2016; (Wright et al., 2018)). In addition, the combination of AO and MI offers performers a deeper mental process (Eaves et al., 2016). First, AO provides performers’ visual information regarding how motor skill is executed, the position of the trunk, foot stance, timing of contacting the objects, arms, and feet movement. So performers can direct their attention to the key point of the movement. On the other hand, MI provides kinesthetic sensations in muscles and joints during motor imagery (Wright, Frank & Bruton, 2021). MI strengthens performers’ memories on trunk and limbs’ angles while performing, or manipulates sporting tools with appropriate speed and force. Especially, we adopted Holmes and Collins’ (2001) PETTLEP approach to engaging in motor imagery. This approach allows participants to integrate all elements of motor skills including physical, environmental, task-specific, timing, learning, emotional, and perspective into the mental simulation. The double mental process of AO and MI is considered to be better than doing AO or MI separately (Eaves et al., 2016).

The other finding that needs discussion is the effect of a different combination of AOMI intervention on golf putting retention tests. At the first and second retention tests, we found all AOMI groups had enhanced performance. The results extend past research (i.e., McNeill et al., 2020; Romano-Smith, Wright & Wakefield, 2018; Romano-Smith et al., 2019; Sun et al., 2016). That is, the AOMI is effective not only in improving immediate golf putting performance but that improved performance was maintained in retention tests. Generally, sport coaches or performance instructors rely on physical practice to improve motor performance. Our findings suggest that AOMI is another approach to enhance motor skill learning.

The effects of S-A-AOMI on retention are worthy of further discussion. Specifically, the mean posttest scores of the A-S-AOMI group were higher than A-AOMI and S-AOMI (99.82; 96.42; and 85.27 respectively). The post-hoc comparison found there is no difference between S-A-AOMI and A-AOMI, but A-S-AOMI scored better than S-AOMI. Di Rienzo et al. (2019) contended that motor imagery is a top-down cognitive process whereas action observation is a bottom-up cognitive process. For an inexperienced learner of mental training, the simultaneous involvement of two cognitive processes such as S-AOMI may mutually conflict and increase the mental burden. Also, Bruton et al. (2020) indicated that using two psychological techniques simultaneously is vulnerable to variations in attention, and reduces the capacity to control attention. This may explain why the S-AOMI group had inferior performance in the retention tests compared to the other two groups. We suggest future studies may examine how S-AOMI influences motor tasks by sampling experienced and inexperienced learners.

In sum, our results outlined that the combination of AOMI has its advantages in motor skill performance and learning. According to Emerson et al. (2022), both action observation and motor imagery can be generated in the brain by a higher order cognitive process. However, when combining AOMI together, in the left prefrontal cortex, the cerebral oxygenation was greater than doing AO or MI alone. This index explains that AOMI activates more neural involvement and functioning. Thus, our research indirectly supports Emerson et al.’s (2022) hypothesis and advances our knowledge in this line of scientific endeavors.

Limitations and future suggestions

Despite these significant and unique findings, there are several limitations in the present study. First, because our participants were all college students whether our results can be applied to other populations such as adolescents, older adults, or patients needs further examination. Further, the golf putting in our study was on artificial turf that runs faster than natural turf. Thus, whether our results can be applied to performance on a real golf course needs further examination. In addition, there are various motor tasks in sports such as continuous (e.g., running, swimming, gymnastic, and golf putting in this study) and discrete (e.g., basketball shooting, archery, dart-throwing), whether our results can be extended to these motor tasks need further examined. Furthermore, during the A-AOMI intervention, although we asked participants not to imagine the action during the AO phase, it is not so easy to control participants unconsciously to imagine the action at the same time. We only asked them to observe the videos carefully and memorize all components of successful putting. To avoid the so-called ”white bear effect (Wegner & Schneider, 2003)” we even didn’t ask them NOT to imagine it. It is one of the limitations that might be happened in the AOMI study. Moreover, past neuroimaging studies indicated that AOMI increases motor cortical involvement compared to AO or MI alone (Ruffieux et al., 2018; Taube et al., 2015). Because we did not measure the neuro activities of S-AOMI, A-AOMI, and A-S-AOMI during the intervention, we should not conclude that S-AOMI is less involved in motor cortical. Finally, Paivio (1985) contended that motivation affects the effectiveness of imagery interventions on motor performance. In our study, the control group only arranged to read a golf celebrity’s story. Such experimental control might result in low motivation compared to experimental groups. To reduce such effects, future studies may use more motivating control conditions that do not involve mental simulation processes.

Applications

We suggest PE teachers, coaches, or performance instructors apply different types of AOMI plus physical training because past research indicates that combining physical training and mental training brings the best results (Afrouzeh, 2015; Battaglia et al., 2014; Pocock et al., 2019). Further, to gain optimal effects from AOMI training, we suggest that PE teachers, coaches, or performance instructors adopt a personal and visualized PETTLEP approach in the MI training. Recent PETTLEP suggests that the personal and visualized PETTLEP approach not only enhances learners’ motivation in imagery training but also improves motor performance (Lu et al., 2020). Furthermore, to those who are not so familiar with mental training, we suggest not using S-AOMI in mental training in order to avoid cognitive conflict and increase the mental burden.

Conclusions

The combination of action observation and motor imagery has received attention from researchers. To advance our knowledge of AOMI combinations, we included a progressive A-S-AOMI approach in our study. Initial results indicated that this progressive A-S-AOMI has similar effects on motor task performance as traditional A-AOMI and S-AOMI. As Emerson et al. (2022) contended when combining AOMI together, it involves more neural activities and functioning in the brain so it can facilitate motor skill performance and learning. Future study is needed to examine the optimal combination of AOMI on motor tasks with varied skill levels participants and different motor tasks.

Supplemental Information

Supplemental Information 1 The raw data

Uses SPSS to enter all experimental data

Click here for additional data file.

Supplemental Information 2 The codebook in this study

Click here for additional data file.

Supplemental Information 3 Imagery script with the contents of PETTLEP imagery

Click here for additional data file.

Additional Information and Declarations

Competing Interests

Author Contributions

Human Ethics

Data Availability

Frank J. Lu is an Academic Editor for PeerJ. The remaining authors declare that they have no competing interests.

Chi-Hsian Lin conceived and designed the experiments, performed the experiments, analyzed the data, prepared figures and/or tables, authored or reviewed drafts of the paper, research fund application, and approved the final draft.

Frank J.H. Lu conceived and designed the experiments, authored or reviewed drafts of the paper, and approved the final draft.

Diane L. Gill analyzed the data, prepared figures and/or tables, authored or reviewed drafts of the paper, and approved the final draft.

Ken Shih-Kuei Huang analyzed the data, prepared figures and/or tables, and approved the final draft.

Shu-Ching Wu performed the experiments, prepared figures and/or tables, and approved the final draft.

Yi-Hsiang Chiu performed the experiments, analyzed the data, prepared figures and/or tables, and approved the final draft.

The following information was supplied relating to ethical approvals (i.e., approving body and any reference numbers):

The Ethical Review Board, National Taiwan University approved the study (NTU-REC (No. 201903ES019).

The following information was supplied regarding data availability:

The raw data is available in the Supplemental File.

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
