# Peer review of "Combinations of action observation and motor imagery on golf putting’s performance"

_PeerJ, doi:10.7717/peerj.13432_

## Round 0.1 · original submission · Major Revisions

· Academic Editor

Major Revisions

Dear Authors,

Two experts in the field reviewed your manuscript and identified several major concerns you should consider in your revision process.

·

Basic reporting

The manuscript reports a result of a comparison of different action observation and motor imagery and intervention combined both techniques to demonstrate the beneficial effects on golf putting performance. Three strategies were used to differentiate the AO and MI, namely, arranging the participants into either S-AOMI group (Simultaneous-AOMI), A-AOMI group (Alternate-AOMI), and A-S-AOMI group (Synthesis-AOMI). The results suggested that, after six weeks of intervention, the participants in the S-AOMI, A-AOMI, and A-S-AOMI groups performed better in the putting task in the post-test when compared to the pre-test. The superior performance was also found in comparing the MI-AO combined groups to the counterparts in the control group. This manuscript is reasonably clear and concise, and the authors show a revealing command of future studies. The authors have done a great job of providing an informative and meaningful addition to the literature of AO and MI in sport psychology.
However, there are several changes that the authors are encouraged to revise to elevate the overall contribution of the paper to this research field.

Experimental design

General comments:
1. Firstly, the authors are recommended to add more information regarding the rationale of conducting the current study, especially why is the rationale behind comparing three different kinds of AO-MI combined tasks and why chose golf putting performance as the designed motor task.
2. Please make the introduction more concise. The authors may want to streamline the theoretical background in the introduction section to elevate the conciseness. One helpful suggestion might be moving some information from the introduction to the discussion to balance the overall information flow.
3. The abstract requires a conclusive implication at the end. It would be nice to see how the authors elucidate the overall findings in the abstract.
4. The overall findings in the current study may require further discussions in sport psychology. For instance, how does the AO/MI relate to motor performance, though which kinds of possible mechanisms? Does the AO/MI approach alter the attentional focus?

Validity of the findings

Specific comments:
1. Line 109, is investigating the optimal order of AO and MI the primary focus in this study? However, the relevant studies cited in this paragraph did not fairly match this point. The authors may want to collect and specify more evidence surrounding the optimal order of AO and MI.
2. Line 115, please specify the type of participants from the study of Romano-Smith. It would make a more precise point to establish the previous findings.
3. Line 123, a more robust rationale is expected than merely comparing the S-AOMI or A-AOMI difference in another motor task. The readers might wonder why the golf putting task set a different tone from the dart-throwing task when taking S-AOMI or A-AOMI into consideration.
4. Line 129, again, it might be nice to explain why the authors chose the golf putting task before introducing it.
5. Line 147, did the gender balance exist in all groups?
6. Line 177, the PETTLEP approach. The authors may want to add additional information regarding PETTLEP approach. Furthermore, why chose this approach?
7. Line 188, the readers might be curious about why using Stimulus-Response Training here. Add more information concerning why and how this training connected to the current study is encouraged.
8. Line 224, is there any supportive evidence to the retention test after one week from the post-test? Moreover, any deliberate control was adopted to ensure the participants did not continuously practice the golf putting?
9. Line 234, the Research Tools section seems a bit redundant. Many pieces of information can be integrated into either the study design section or the experiment procedure section.
10. Line 272, the skill performance scoring. Did all the participants perceive a similar difficulty when performing the putting task? 180 cm from the hole might be a good start for the novices. However, some might feel hard to put the ball into the hall because the distance might not be the factor to introduce the difficulties for novices. Furthermore, was the artificial turf inclined or having any curves?
11. Line 279, the readers might wonder why scoring 1 and 2 are two different stories. Was the putting performance accessed to exemplify the accuracy or something else?
12. Line 303, please keep the stats in a consistent format throughout the manuscript.
13. Line 356. The explanation should be careful as the studies cited above do not provide collective support to this interpretation. That is, Wang et al. (2020) made the comparison on elite golfers, not novices. These findings may not follow the following findings, which suggested that enhancing motor cortical activity in the brain resulted from AO and MI (Avanzino et al., 2011; Jeannerod, 2001; Tani et al., 2018). Hence, the enhanced motor performance after the intervention may require further evidence to support this claim.

·

Basic reporting

Overview
This is a neat study which addresses a relevant and contemporary topic in this field of research. I think this work makes a useful contribution to the field because, and as the authors state, previous AO+MI golf studies have not included a retention test, and you did. The authors employed a between group pre- post – retention (1 and 2) design to compare three types of mental practice against each other and a no practice control group. At the important second retention test, synchronous AO+MI does not perform as well as asynchronous AO+MI or a group where these two AO+MI instructions are combined. I think the main components of a good study are reported in the manuscript, but I recommend the authors consider the points I raise below as a means to improve the write up further prior to publication.


Introduction
I don’t think it is necessary to recapitulate several dated theories at the start of your intro (line 57 – 76). You don’t test these theories directly and some are not well supported by empirical work (and you don’t include an evaluation of the available evidence supporting each theory. My suggestion would be to remove at least the first two (psychoneuromuscular and symbolic learning theory). I would briefly add some empirical support for bioinformational theory – please see earlier work e.g., by Jenn Cummings (University of Birmingham)

For further clarity and improve readability in the main terms you use, I would recommend using:
- Synchronous AO+MI (instead of S-AOMI)
- Asynchronous AO+MI (instead of A-AOMI)
- I don’t think the term ‘synthesised’ really captures the methodology you used in the third condition. Instead of synthesised AOMI (also quite similar to my new suggestion of synchronous) you could consider terms like: graduated or progressive AO+MI

Line 57: imagery works… to achieve what? Please expand
Line 83: needs a reference, e.g.:
Rizzolatti, G., Maddalena, F.D., Arturo, N., Gatti, R. & Pietro, A. (2021). The role of mirror mechanism in the recovery, maintenance, and acquisition of motor abilities. Neuroscience and Biobehavioral Reviews.

Line 102: the word ‘then’ implies that the two processes (AO and MI) occur in a sequential fashion, when this is not the meaning you intend – consider replacing, e.g., with ‘and imagining’ … ‘either / or‘ ?

Line 128: I would not say that you extend Romano-Smith’s study, as you use a completely different task. Please re-phrase

To support hypothesis 2, I would suggest adding a little extra detail on the rationale behind the order of presentation used in A-S-AOMI… namely that you place alternating first – do you presume this is less demanding and more helpful in early learning than S-AOMI? If so, two studies support the idea that extra neurocognitive demands are involved in the latter:

- Emerson, J. R., Scott, M. W., Van Schaik, P., Kenny, R. P. W., & Eaves, D.L. (2022). A neural signature for combined action observation and motor imagery? An fNIRS study into prefrontal activation, automatic imitation, and self-other perceptions. Brain and Behaviour. e2407.

- Eaves, D. L., Behmer, L. P., & Vogt, S. (2016b). EEG and behavioural correlates of different forms of motor imagery during action observation in rhythmical actions. Brain and Cognition, 106, 90-103.


the writing style is mainly approriate throughout - take care to avoid small typos throughout

Experimental design

methodology
Line 176 – please include the sample imagery script in your supplementary materials and make it clear how this follows the pettlep principles in context. Was the imagery assessment used as a screening tool for inclusion in the study? If so, please describe the cut off for inclusion?

Line 186 do you mean 6 blocks of trials comprising 5 trials each (n= 30)?

Line 207. Please add more detail to the imagery instruction component of the S-AOMI condition – what specifically did you ask them to imagine during the video, for e.g., the kinaesthetic physical sensation of effort and force? Did this include trunk rotation, and body/head position and also did the imagery end when the visual component related to ball trajectory in the video – did they see the ball go in the hole on each trial and if so, what were they imagining at that point?

Line 209: what were the observation instructions exactly? Did you say just watch, or did you direct them to observe the same components of action described in the imagery? Importantly – did you take any steps to ask participants not to undertake imagery in the AO phase of each trial? Please state in the paper if you believe this could be an explanation for the main findings. This point also applies to A-S-AOMI.
Please state the total number of AO and MI trials in A-S-AOMI as well as the total number of AOMI trials in this group. It is important to be able to compare the number of trials across all groups… I presume all groups had 30 trials per intervention session, but it will be good to see the break down (to allow for assessing dose response

Please make it clear in the methods section that the intervention comprised purely of mental practice in the absence of physical practice throughout the intervention period (not withstanding the physical execution required at the repeat baseline testing points

Pre test and post test – please state the number of physical trials undertake at each of these two time points. For retention test please identify each time point as described in the results section (retention test 1 and retention test 2)

Validity of the findings

Results
Table 2. it is not good to present the same data twice. To avoid this, please add the SD bars on to figure 3 and remove table 2.
From your stats reports it is not clear if there is a significant difference between the four groups at the initial pre-test – please report… looking at figure 3 it seems there is an imbalance in the scores at this time point (despite the fact you have not added comparison bars to highlight sig difs.
Please report the post hoc analysis to follow up on the main effect of test (i.e., time)
It is also difficult to see in your stats what the results are for the comparisons within each group – e.g., do they all increase performance over time to the same extent?
Given the unstated analysis of differences between groups at the pre-test, please could you explain why you have opted not to run an analysis of co-variance on the data, using the base line score as the co-variant? This will be necessary if there are indeed differences between groups at the baseline
Discussion
Limitations: please comment on the possible confound of spontaneous MI during the so-called AO only phases of the methodology – what actions did you take to reduce this possibility and what impact might this potential confound have on your data?

How do you account for the finding that the S-AOMI group performed worse than the A-AOMI group, given the large number of neuro imaging studies that show the neural involvement is very different between these two instructions? The general pattern of results is that AOMI significantly increases motor cortical involvement compared to AO or MI alone, and this is assumed to be a good thing for motor learning see:

Ruffieux, J., Mouthon, A., Keller, M., Mouthon, M., Annoni, J.M. & Taube, W. (2018). Balance training reduces brain activity during motor simulation of a challenging balance task in older adults: an fMRI study. Frontiers in Behavioral Neuroscience, 12, 10.

Taube, W., Mouthon, M., Leukel, C., Hoogewood, H., Annoni, J., & Keller, M. (2015). Brain activity during observation and motor imagery of different balance tasks: An fMRI study. Cortex, 64, 102-114.

The conclusion starting in line 408 is wrong or at least too blunt – you did find sig differences between these three conditions at both retention tests. Clearly the second delayed retention test is the most crucial time point to highlight in your discussion section for evaluating your findings. Here you show:
A-AOMI and A-S-AOMI > control group
A-S-AOMI > S-AOMI
I think the latter finding specifically speaks to the advantage offered early in learning from asynchronous AO and then MI… this also suggests that your task was suitably novel / challenging for your participants… it could be anticipated (based on these results) that as motor skill develops, performers can graduate toward using S-AOMI and then perhaps favour MI alone, but this remains to be tested at advanced skill levels

Additional comments

.

---

## Round 0.2 · Minor Revisions

· Academic Editor

Minor Revisions

Dear Authors, two experts in the field revised your manuscript founding some minor issues you should consider while revising the manuscript.

·

Basic reporting

The revised manuscript is more concise and clear than the previous version. My compliments to the authors.
Just a few revisions should take into account before the final decision.

Line 136. The study contained AOMI (S-AOMI) and AOMI (A-AOMI) approaches. The readers might wonder how these two approaches affect the psychological and cognitive processes after the training. More supportive inferences might be helpful to explain the results which using AOMI (S-AOMI) and AOMI (A-AOMI).

Figure 2 looks like it can be upgraded by using more proper software to display the results. Please reform Figure 2.

The study design. The authors are recommended to implement a flowchart or framework to showcase the study design. This addition would make the readers understand the design of the current study more easily.

Experimental design

No further comments on this part.

Validity of the findings

Line 519-520. I would expect an informative indication of the main results here...before simply introducing the optimal combination of AOMI for future study. Again, similar to the previous comments, the results should be linked with the inference to the psychological or cognitive processing of motor performance.

Additional comments

Taken together, the authors did an excellent job of elevating the overall quality of the current manuscript. However, based on the spirit of advancing the boundary of AO/MI fields in sports, more theoretical evidence and interpretation should be provided and added to explain the findings.

·

Basic reporting

I am happy that the authors have revised the manuscript in accordance with and to address my initial concerns

Experimental design

.

Validity of the findings

.

Additional comments

.

---

## Round 0.3 · accepted · Accept

· Academic Editor

Accept

Dear Authors, one expert in the filed revised your manuscript, suggesting it for publication.

·

Basic reporting

I have no further issues with the manuscript and I recommend the manuscript for PeerJ publication.

Experimental design

This parts sounds good to me.

Validity of the findings

The authors have revised our comments accordingly.